# Response of salt stress resistance in highland barley (*Hordeum vulgare L. var. nudum*) through phenylpropane metabolic pathway

ZhengLian Xue☯, BingSheng Wang☯, ChangYu Qu, MengDie Tao, Zhou Wang, GuoQiang Zhang, Ming Zhao🅳*, ShiGuang Zhao*

Anhui Engineering Laboratory for Industrial Microbiology Molecular Breeding, College of Biological and Food Engineering, Anhui Polytechnic University, Wuhu, China

☯ These authors contributed equally to this work.
* zhaom@mail.ahpu.edu.cn (MZ); zhaoshiguang@ahpu.edu.cn (SZ)

**Data Availability Statement:** All the RNA-seq data generated in this research was deposited in the Sequence Read Archive database (www.ncbi.nlm.

## Abstract

Highland barley (*Hordeum vulgare L. var. nudum*) is a grain crop that grows on the plateau under poor and high salt conditions. Therefore, to cultivate high-quality highland barley varieties, it is necessary to study the molecular mechanism of strong resistance in highland barley, which has not been clearly explained. In this study, a high concentration of NaCl (240 mmol/L), simulating the unfavorable environment, was used to spray the treated highland barley seeds. Transcriptomic analysis revealed that the expression of more than 8,000 genes in highland barley seed cells was significantly altered, suggesting that the metabolic landscape of the cells was deeply changed under salt stress. Through the KEGG analysis, the phenylpropane metabolic pathway was significantly up-regulated under salt stress, resulting in the accumulation of polyphenols, flavonoids, and lignin, the metabolites for improving the stress resistance of highland barley seed cells, being increased 2.71, 1.22, and 1.17 times, respectively. This study discovered that the phenylpropane metabolic pathway was a significant step forward in understanding the stress resistance of highland barley, and provided new insights into the roles of molecular mechanisms in plant defense.

## Introduction

Highland barley (Hordeum vulgare L. var. nudum), barley genus [1], is mainly distributed in northwest and southwest China's highland areas. It is not only the main grain for Tibetan residents in China, but also the primary raw material for producing of beer, medicine, and health products [2]. Compared with conventional barley varieties, highland barley has the properties of high protein, high dietary fiber, and low fat structure [3], which meet the requirements of modern people for food. Therefore, highland barley has a broad application prospect in the functional food processing industry. Presently, research on highland barley mainly focuses on determining nutritional composition, extracting active ingredients, and analysing [4, 5]. For example, Dang [6] studied the content and antioxidant capacity of β-glucan, melatonin, total phenols and flavonoids in highland barley seed during the germination stage, and analyzed the antioxidant capacity of different tissues of highland barley. Therefore, the study of the

nih.gov/geo, accessed on 16 Nov 2022) at NCBI (National Center for Biotechnology Information) under accession number: PRJNA902282.

**Funding:** This research was funded by a grant Tibet Key Research and Development Program Program (Program no. XZ202001ZY0040N), a grant (2022jc19) from Science and Technology Plan Project of Wuhu, and a grant (B2018-03) from Xuancheng Industrial Technology Research Institute Support Project. The funders had no role in study design, data collection and analysis, decision to publish, or preparation of the manuscript.

**Competing interests:** The authors have declared that no competing interests exist.

resistance of highland barley to the unique living environment is significant for cultivating high-quality highland barley varieties and improving the resistance of different plant species.

Salt stress is an important means of studying the stress resistance of plants [7]. Soil salinization and soil osmotic pressure can be increased by the changes in the concentration of $Na^+$ and $K^+$, thus causing physiological drought and inhibiting normal plants growth [8]. Excessive salt ions accumulate in the plant tissues, resulting in ion antagonism, obstruction and destruction of normal physiological metabolism, and plant malformation or death [9]. In addition, the redox balance in plant cells is destroyed under salt stress. Therefore, to effectively reduce the damage of reactive oxygen species, cells need to regulate the expression of a protective enzyme system to form a self-defense system [10]. In recent years, omics technology has developed rapidly, and it is wildly used to excavate stress tolerance genes in plants, especially in the analysis of the molecular mechanism of salt tolerance of plant cells [11–13]. However, the results of the molecular mechanism of salt stress in highland barley are still unsatisfactory [14]. Therefore, the analysis of the molecular mechanism of salt stress in highland barley is very important for researching and cultivating plant varieties with high-stress resistance.

Researchers mainly use transcriptomic strategies to study the differences in gene expression levels of highland barley under salt stress to excavate stress-related genes. For example, Lai [15] utilized transcriptomics and proteomics strategies to analyze the molecular mechanism of salt stress response in highland barley seed cells during germination stage. They found that highland barley seeds could germinate normally under salt stress conditions and detected many genes related to salt response, which initially revealed the molecular mechanism of highland barley salt tolerance. However, the mechanism of salt stress response in related pathways remains unclear. Therefore, it is urgent to investigate further the molecular mechanism of salt stress in highland barley. Recently, the phenylpropane metabolic pathway, which produces polyphenols, flavonoids, lignin, and other secondary metabolites, has been reported as an important branch for the regulation of stress resistance in Pontederia cordata [16], Walnut [17], and watermelons [18]. These metabolites can effectively decrease the accumulation of reactive oxygen in plant cells and prevent cell morphology, structure, and physiological metabolic function destruction. Therefore, the phenylpropane metabolic pathway is closely related to the stress resistance of highland barley [19]. In this study, NaCl was used to simulate the salt stress environment to stimulate the seeds of highland barley during the germination stage. Different periods transcriptomics were used to analyze the molecular mechanism of highland barley stress resistance (S1 Fig).

## Materials and methods

### Materials

Tibet Qing 2000 was provided by the Academy of Agriculture and Animal Husbandry Sciences of Tibet Autonomous Region. Gallic acid standard, rutin standard, ferulic acid standard, caffeic acid standard, Folin-Ciocalteu's phenol, and lignin content detection kit were purchased from Sangon Biotech. RNA Easy Fast RNA extraction kit for plant tissue, FastKing cDNA first strand synthesis kit, and RealUniversal color fluorescent quantitative premix reagent were purchased from TIANGEN (Beijing, China). qRT-PCR primers were synthesized at GenScript (Nanjing, China).

### Plant growth conditions

Highland barley seeds full of grains and no mechanical damage were selected and soaked in 1 g/L NaClO for 5 min. Then, highland barley seeds were cleaned off and soaked for 10 h at 25°C. About 20 g of soaked seeds are placed in each marked seedling box. The treated seeds

were germinated and grown in the intelligent artificial climate chamber (relative humidity of 80%, 12 h light/12 h dark, 25˚C). Highland barley has its unique tolerance to adversity and is very suitable for growing on saline soil as a food crop. Seeds were sprayed with 240 mmol/L of NaCl and sterilized water daily to collect highland barley seedlings on 0, 1, 3, 5, and 7 days.

## Preparation of highland barley extract solution

Referring to Aguilera's method with slight modifications [20], the ultrasonic assisted extraction method can dissolve polyphenols, flavonoids and other substances contained in plants in methanol as completely as possible. Approximately 0.2 g of highland barley freeze-dried powder was weighed and added into 5 mL 50% methanol solution for shock mixing. The solution was left in darkness at 4˚C for 16–18 hours, then ultrasonic crushed in the water bath for 30 min. Finally, the treated solution was centrifuged at 10,000 r/min for 15 min at 4˚C, and the supernatant was filtered through the 0.45 μm microporous filter membrane and stored in the brown flask.

## Determination of total polyphenol content

The gallic acid standard solution (10 mg of gallic acid standard was weighed and dissolved in distilled water to 50 mL) of 0, 10, 20, 30, 40, and 50 μL were added into the well plate, respectively, mixing with 10 μL Folin-phenol reagent or static reaction for 6 minutes. 100 μL of 7% (W/V) $Na_2CO_3$ was added to the solution, distilled water was added and the solution volume was adjusted to 290 μL. Absorbance was measured at 760 nm and the standard curve was drawn. 50 μL highland barley extract solution was added into the well plate and analyzed with the above method.

## Determination of total flavonoids content

The rutin standard solution (10 mg of rutin standard was weighed and dissolved in 80% ethanol to 50 mL) of 0, 10, 20, 30, 40, and 50 μL were added to the well plate, mixing with 20 μL of 5% (W/V) $Na_2CO_3$ solution for a static response for 6 min. Then, 200 μL 10% (W/V) $Al_2(NO_3)_3$ was added for a static reaction for 6 min. 200 μL 4% (W/V) NaOH was added in the end and the mixture solution was reacted for 10 min at room temperature for 15 minutes in the dark. Absorption was measured at 510 nm and the standard curve was drawn. 50 μL highland barley extract solution was added to the well plate and analyzed with the above method.

## Determination of lignin content

The lignin content was determined according to the characteristic absorption peak at 280 nm after acetylation of the phenolic hydroxyl group in the lignin. The operation procedure followed the instructions of the lignin content detection kit.

## HPLC analysis of ferulic acid and caffeic acid

The content of ferulic acid in highland barley extract solution was determined by C18 (4.6 mm×250 mm, 5 μm) column Arslan [21]. The mobile phase was methanol (A): 0.1% (V/V) phosphoric acid solution (B) (20: 80). The detection wavelength was 320 nm; the flow rate was 1.0 mL/min, the column temperature was 30˚C and the sample size was 10 μL. Ferulic acid standard solution (5.01 mg of standard ferulic acid was dissolved in 10 mL of methanol solution. 1 mL of the above solution was added into a 100 mL beaker. Methanol was added to the scale line and mixed to obtain 5.01 μg/mL of ferulic acid solution) was diluted to various concentrations. Then the standard curve was drawn according to peak area and peak time. The

content of caffeic acid in highland barley extract solution was determined by C18 (4.6 mm×250 mm, 5 μm) column Wang [22]. The mobile phase was acetonitrile (A): 0.1% phosphoric acid solution (B) (10: 90); detection wavelength was 320 nm; flow rate was 1.0 mL/min; column temperature was 35˚C; sample size was 20 μL. Caffeic acid standard solution (21.72 mg ferulic acid standard was dissolved in 10 mL methanol solution. 1 mL of the above solution was added into a 100 mL beaker. Methanol was added to the scale line and mixed to obtain 21.72 μg/mL) it was diluted to various concentrations and then the caffeic acid standard curve was drawn. Highland barley extract solution was injected after filtering through a 0.22 μm microporous filter membrane.

## Transcriptomic sequencing

The germination and cultivation process of highland barley seeds were the same as above. Highland barley seedlings with even growth at 0, 1, 3, 5, and 7 d were snap-frozen in liquid nitrogen and stored at -80˚C. Plant RNA was extracted and reverse transcribed to obtain cDNA. Three biological repeats were set for each sample. These libraries were sequenced by paired end (PE) sequencing based on the Illumina sequencing platform of Boser Biotech at Personalbio (Shanghai, China). KEGG enrichment analysis of differential genes in transcriptome data was performed by clusterprofiler.

## qRT-PCR

Total RNA was extracted from highland barley using the RNA Easy Fast RNA Extraction kit purchased from TIANGEN (Beijing) according to the manufacturer's instructions. *gapdh* was used as the housekeeping gene, qRT-PCR was performed using a FastKing cDNA first strand synthesis kit and RealUniversal color fluorescent quantitative premix reagent purchased from Tiangen (Beijing, China) according to the manufacturer's instructions. For each PCR reaction, 10 μL of 2×RealUniversal PreMix, 0.6 μL of forward and reverse primers (S5 Table), 1.5 μL of cDNA sample, and 7.3 μL of RNase-Free ddH$_2$O were mixed, and PCR conditions were as follows: 95˚C for 15 min, followed by 40 cycles of 95˚C for 10 s and 60˚C for 30 s. The result was analyzed with the $2^{-\Delta\Delta Ct}$ method and averaged with triplicates.

## Results

### Analysis of antioxidants in highland barley seeds under salt stress

Salt stress will cause plant cells to produce a large amount of free oxygen, resulting in oxidative damage to plant cells [23]. Polyphenols and flavonoids were demonstrated as strong antioxidants, which could effectively remove oxygen free radicals in plant cells, thus protecting cells from the damage of cell-free toxicity [24]. Therefore, in this study, the content of total polyphenols and flavonoids in highland barley seeds during seed germination stage under salt stress and nonsalt stress environment was first detected. The contents of total polyphenols and total flavonoids in highland barley reached the maximum under the condition of 240 mmol/L NaCl (S2 Fig). As shown in Fig 1a and 1b, the maximum accumulation of polyphenols and flavonoids was 8.38±0.36 mg/g and 1.982±0.08 mg/g on the third day (S1 Table), respectively, reaching 2.71 and 1.22 times of the accumulation of polyphenols and flavonoids in highland barley seed cells under normal conditions. These results illustrated that polyphenols and flavonoids content of highland barley seed cells could be efficiently increased under salt stress, thus resulting in the rapid removal of free oxygen in highland barley seed cells.

It is widely known that ferulic acid and caffeic acid are two primary polyphenols in the plant [25, 26]. Therefore, ferulic acid and caffeic acid content in the highland barley seeds was

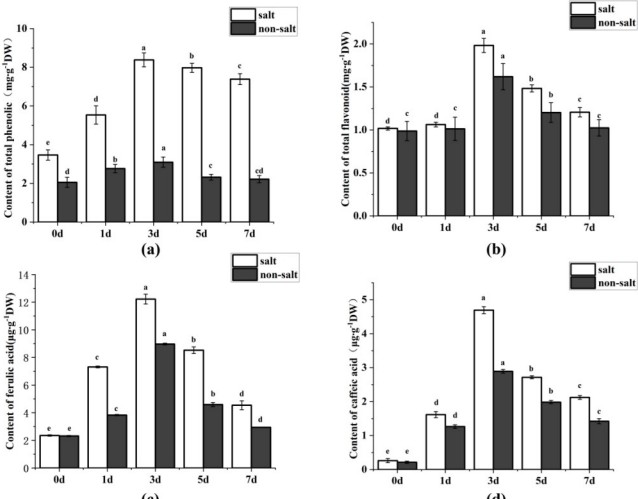

**Fig 1. Content analysis of highland barley seeds under salt stress.** The contents of total polyphenols and total flavonoids changed on different germination days (0 day, 1 day, 3 day, 5 day, and 7 day) under salt stress. (**a**) The content of total polyphenols on different germination days. (**b**) The contents of total flavonoids changes on different germination days. (**c**) The contents of ferulic acid content on different germination days. (**d**) The contents of caffeic acid on different germination days. Error bars show the standard deviation from three independent experiments.

also detected. The results exhibited that the maximum accumulation of ferulic acid and caffeic acid was 12.22±1.35 µg/g and 4.69±0.95 µg/g on the third day, respectively, reaching 1.36 and 1.62 times the accumulation of ferulic acid and caffeic acid in highland barley seed cells under non-salt environment. These results implied that ferulic acid and caffeic acid would probably be the most important contents in highland barley for resisting disadvantage environments.

Besides the antioxidants, physical factors would also enhance the mechanical strength of highland barley seeds, thus resulting in protection against the external environment. Plant tissues contain a large amount of lignin, which is an important component to maintain the extremely high hardness of plants and protect tissues from corrosion [27]. Therefore, lignin accumulation during germination of highland barley seed was also detected in this study. As shown in Fig 2, lignin content increased with the germination time of highland barley seed and reached the highest accumulation on the seventh day. In addition, lignin synthesis capacity was significantly improved under salt stress, reaching 378.64±0.91 mg/g on the seventh day, which was 1.17 times the seed content without salt stress. These results indicated that highland barley seeds might resist the external environment by enhancing lignin synthesis in the process of stress resistance.

## Transcriptomic analysis of gene expression levels of highland barley seed cells under salt stress

Under salt stress treatment, the synthesis capacity of antioxidants polyphenols and flavonoids of highland barley seeds, as well as lignin, were significantly improved during the germination stage, meaning that the transcription level of genes for the regulation and biosynthesis of these metabolites was significantly changed under salt stress. To confirm this hypothesis, transcriptomic profiles of highland barley seeds germinating on the third and fifth days under salt stress were evaluated compared to seeds under non-salt stress. A total of 100 Gb of clean, high quality data was obtained using Illumina HiSeq platform. The ratio of useful reads varied from 91% to

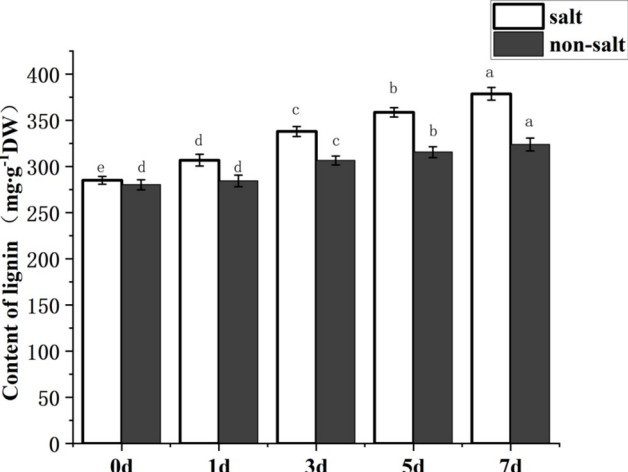

**Fig 2. Content analysis of lignin in highland barley seeds on different germination days.** Error bars show the standard deviation from three independent experiments.

95.35% (S2 Table). As shown in Fig 3a and 3b, and S3 Fig, the transcriptional patterns of seeds germinated under salt stress on the third and fifth days exhibited considerable differences from that of seeds germinated under non-salt stress, with 5,220 and 4,788 upregulated genes and 4,293 and 4,565 downregulated genes ($|\log_2 \text{FoldChange}| > 1$; $p < 0.05$), respectively, among about 55,706 total genes. These data clearly demonstrated that the global metabolism of highland barley seed cells underwent tremendous changes under salt stress. Apart from that, to explore the relationship between the changes in genes expression and germination times, the transcriptional patterns of seeds germinated at 1, 3, 5, and 7 days were compared with seeds germinated at 0 days under salt stress. The results showed that more than 20% of the genes in highland barley seed cells were significantly altered during the germination stage under salt stress (Fig 3a and 3d). Except that, the number of up-regulated genes was the largest on the third day during the germination stage, when the accumulation of antioxidants was the maximum. These results illustrated that the metabolism pathways of highland barley seed cells were significantly altered under salt stress, especially during the germination stage after 3 days.

The Venn diagram was used to confirm the relationship between the changed genes. As exhibited in Fig 3c, the transcription levels of 4453 genes and 4295 genes were just changed on the third and fifth day, respectively, illustrating that specific genes' transcription levels differed during different germination stages. Interestingly, the transcription level of about 5058 genes was significantly altered both on the third day and fifth day, indicating that the transcription level of a large number of genes was continuously affected under salt stress.

### Changes analysis of metabolic pathways of highland barley seeds cells under salt stress

After salt stress treatment, the transcription level of more than 20% of the genes in highland barley seed cells was changed. In addition, the accumulation of antioxidants, including polyphenols and flavonoids, and anticorrosive substance, lignin, of highland barley seeds were efficiently increased. Therefore, metabolic pathways in highland barley seed cells should be significantly changed. To investigate this hypothesis, KEGG (Kyoto Encyclopedia of Genes and Genomes) pathway enrichment analysis was used to identify metabolic differences

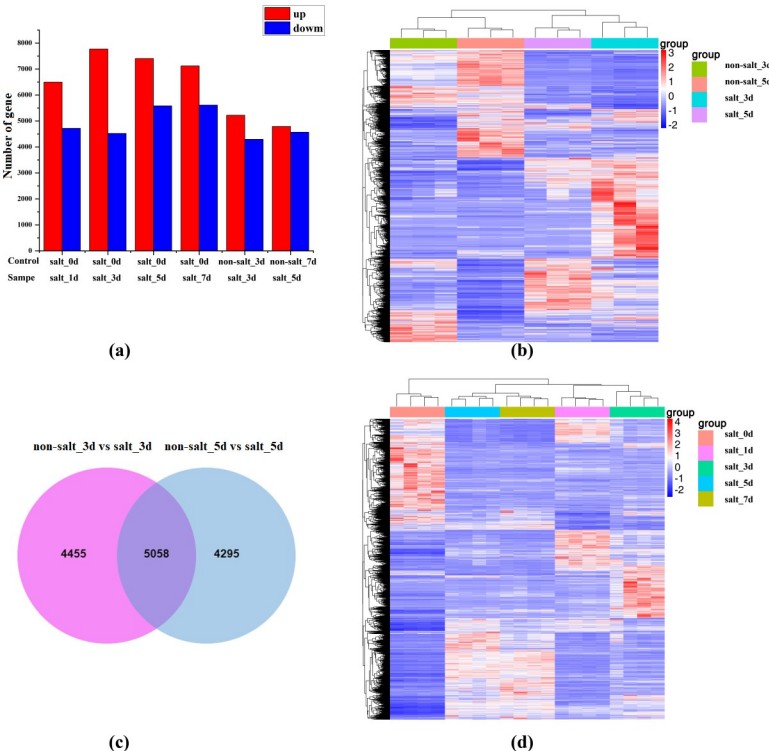

**Fig 3. Transcriptomic profiling of highland barley seeds cells under salt stress.** (**a**) Number of genes significantly (|log$_2$FoldChange| > 1; p < 0.05) differentially expressed in highland barley seed cells during different times of germination stage under salt stress and non-salt stress. (**b**) Heatmap of transcript diversity across highland barley seeds cells during different germination stages (1 day, 3 day, 5 day, and 7 day) under salt stress. (**c**) Venn diagram analysis of highland barley seeds cells during the germination stage (3 day and 5 day) under salt stress and non-salt stress. (**d**) Heatmap of transcript diversity across highland barley seeds cells during the germination stage (3 day and 5 day) under salt stress and non-salt stress. Three biological replicates were performed per sample.

between highland barley seed cells during the germination stage under salt and non-salt stress (Fig 4 and S4 Fig). Changes in ribosomal pathways stood out among metabolic variations. Among 123 genes involved in ribosomal pathways, more than 90 genes exhibited upregulated transcriptional levels on the third day during germination stage (S3 Table), suggesting that profound changes occurred in the global gene translation networks of the highland barley seed cells under salt stress. Interestingly, significant changes took place in the phenylpropane bio-synthesis pathways. Under salt stress, 122 genes and 48 genes in the phenylpropane biosynthe-sis pathway (total 351 genes) were efficiently up-regulated and down-regulated, respectively, during the germination stage (3 day), thus resulting in the improvement of the accumulation of its metabolites, such as polyphenols, flavonoids, and other substances, in highland barley seeds. It is worth noting that significant changes also took place in the glycolysis and TCA cycle pathways. Interestingly, some key genes in the TCA cycle were significantly downregu-lated (3 genes up-regulated and 3 genes down-regulated), while some key genes of the glycoly-sis pathway were upregulated (19 genes up-regulated and 22 genes down-regulated), indicating that additional precursors could be available for the biosynthesis above metabolites in highland barley seeds under salt stress.

The phenylpropanoid metabolic pathway is the core pathway for producing of polyphenols, flavonoids, and other substances, which have been demonstrated as antioxidants in plants to

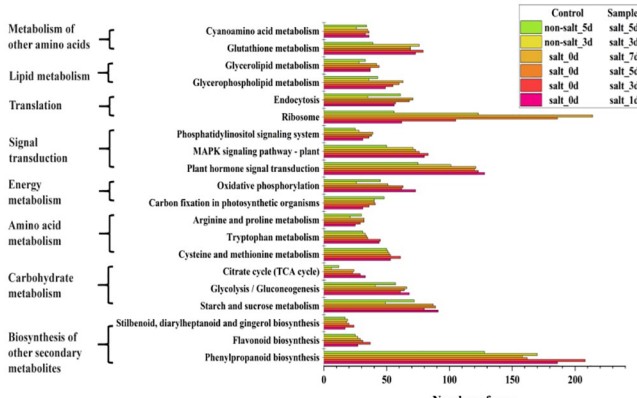

**Fig 4. Functional enrichment analysis of transcripts based on KEGG pathways.** Three biological replicates were performed per condition.

defend against stress. This metabolic pathway starts from phenylalanine, catalyzed by phenylalanine ammonia lyase (PAL) to produce trans-cinnamic acid. In addition, cinnamic acid 4-hydroxylase (C4H) catalyzes trans-cinnamic acid to produce coumaric acid, ferulic acid, erucic acid, caffeic acid, and others [28]. Finally, 4-coumarate-CoA ligase (4CL), cinnamyl-alcohol dehydrogenase (CAD), caffeate 3-O-methyltransferase (COMT), caffeoyl CoA 3-O-methyltransferase (COMT), caffeoyl CoA 3-O-methyltransferase (CCoAOMT), and cinnamoyl-CoA reductase (CCR) work together to synthesize phenylpropanoid metabolites, such as lignin, flavones, and polyphenols. In the analysis of KEGG data (Fig 5, S5 Fig, and S4 Table), we found that more than half of the genes related to the phenylpropane metabolic pathways were significantly up-regulated, while less than 10% were downregulated. These results indicated that highland barley seed cells enhanced the synthesis ability of polyphenolic metabolites via enhancing phenylpropane metabolic pathways under salt stress, thus reducing free oxygen content in cells and defending stress resistance. Interestingly, the phenylpropane metabolic pathway was the most critical pathway associated with the stress resistance of highland barley,

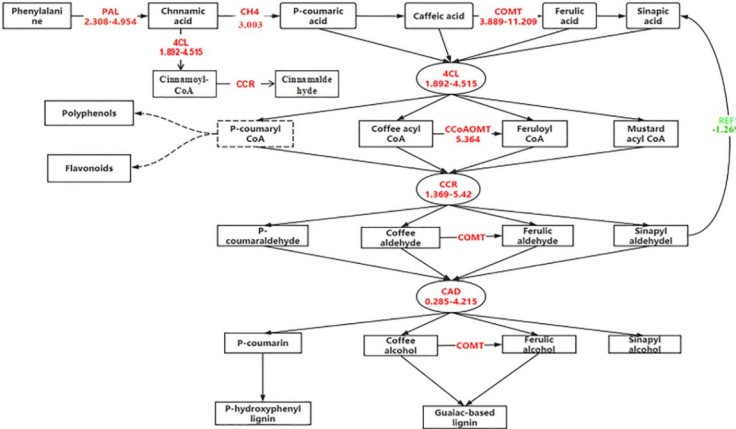

**Fig 5. Functional enrichment analysis of phenylpropanoid metabolic pathway.** Red and green lines represent the upregulated and downregulated pathways, respectively.

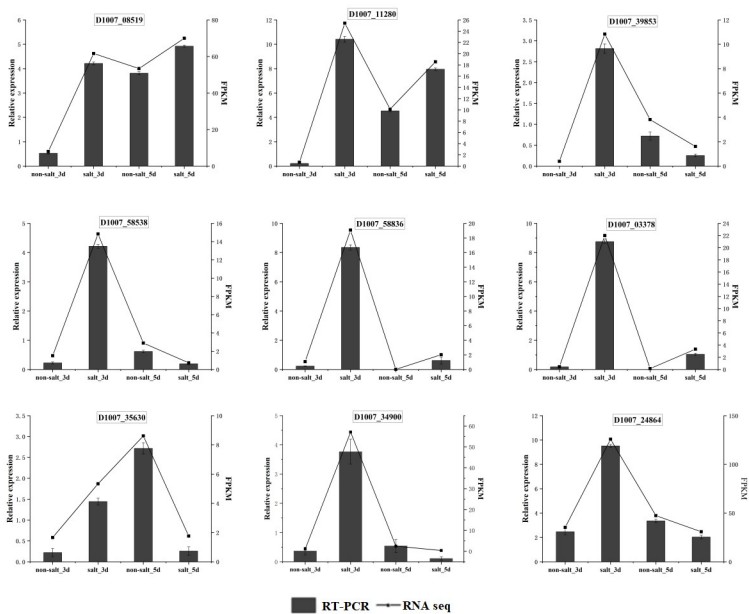

**Fig 6. qRT-PCR verification of the key genes in the phenylpropane metabolic pathway.** The mRNA expression levels were normalized to the expression level of *gapdh*. Error bars show the standard deviation from three independent experiments.

which provided a theoretical basis for studying the resistance characteristics of highland barley and other plant species and the cultivation of resistant strains.

## Verification of the key genes in phenylpropane pathway

In order to further verify the key points of the phenylpropane metabolic pathway in highland barley seed cells in response to salt stress, quantitative reverse transcription polymerase chain reaction (qRT-PCR) was performed to analyze the transcriptional level of each key metabolic gene, including two PAL genes (D1007_08519 and D1007_11280), CH4 gene (D1007_58538), 4CL gene (D1007_39853), COMT gene (D1007_58836), CCoAOMT gene (D1007_35630), CCR gene (D1007_03378 and D1007_34900), and CAD gene (D1007_24864). Comparison of the relative expression levels of these genes evaluated by FPKM and qPCR methods is shown in Fig 6. The results showed that the transcriptional levels of these 9 genes analyzed by qRT-PCR were consistent with that of RNA seq, which verified that the key points of the phenylpropane metabolic pathway were indeed significantly changed under salt stress.

## Discussion

Highland barley is an important grain crop in the plateau area of China. It is rich in nutrients and meets people's demand for a healthy diet [29]. In addition, highland barley has a strong stress resistance, so as to ensure the normal growth of highland barley crops in the adverse environment [30]. Therefore, in recent years, more and more researchers have begun to study the stress resistance mechanism of highland barley, which not only helps to cultivate highland barley crops suitable for different adverse environments, but also provides a reference for the improvement of stress resistance of other crops. However, the current research on stress resistance in highland barley has mainly focused on the content analysis of metabolites [31–33], and gene mining and verification through omic, the work for the identification of metabolic

pathways related to stress resistance is unsatisfactory. This study used the germination of highland barley seeds under salt stress to simulate the unfavorable environment of highland barley crops, and the effective improvement of related genes in the phenylpropane metabolic pathway was located through the transcriptional level of highland barley seeds during different stages under salt stress.

The phenylpropane metabolic pathway is ubiquitous in plants, and its secondary metabolites play an important role in plant growth, development, and defending stress [34]. The biosynthesis of phenylpropane metabolites starts from the common metabolic pathway of phenylpropane. Several branches of phenylpropane metabolism are downstream, including the lignin synthesis pathway, polyphenol synthesis pathway, and flavonoid synthesis pathway. Lignin is the main component of the cell wall [35] that supports the plant upright and allows the xylem to withstand the negative pressure generated during water transport. Under salt stress, the lignification degree of plant root cell wall will increase, which can not only effectively prevent the internal ion absorption of cells, but also enhance the structural rigidity and firmness of conduction tissue and improve the plant salt tolerance [36]. Polyphenols and flavonoids have been demonstrated as strong antioxidants, effectively removing oxygen free radicals in plant cells, thus participating in plant stress resistance. Therefore, the phenylpropane metabolic pathway is probably the most important method for stress resistance in highland barley.

The phenylpropane metabolic pathway has not been reported to be related to stress resistance in highland barley, but it has been studied in other crops [37]. For example, Kiani [38] found that flavonoids in Tartary buckwheat flowers increased significantly after 12 h under salt stress, which may indicate that anthocyanins were synthesized from the corresponding products of salt stress to resist the adverse effects of salt stress [39]. Benincasa [40] found that the content of polyphenols in rapeseed was increased under salt stress, improving plants' resistance ability to salt stress. Therefore, in this study, the barley seeds were treated with 240 mmol/L NaCl, and the total polyphenols, flavonoids, and lignin in highland barley seed cells were measured. It was found that under salt stress, the yield of highland barley seeds polyphenols, flavonoids, and lignin increased by about 50%, 25%, and 20%, respectively (Figs 1 and 2). The results indicated that highland barley seed cells could effectively increase the biosynthesis capacity of polyphenols, flavonoids, and lignin by enhancing the phenylpropane metabolism pathway under salt stress, thus improving the ability of cells to degrade free oxygen generated under adverse conditions and resistance to mechanical forces.

Considering that the biosynthesis capacity of these metabolites was directly according to the transcription level of metabolism pathways, therefore, the transcriptome sequencing method was used to analyze the differences of genes transcription. The results showed that about the transcriptional level of 20% total genes were significantly changed under salt stress (Fig 3), illustrating the global metabolism of highland barley seed cells could be influenced by salt stress. Through the KEGG analysis, we found that about half of the genes in the phenylpropane biosynthesis pathway were significantly changed (Fig 4 and S4 Fig), thus resulting in the improvement of the accumulation of its metabolites, such as polyphenols, flavonoids, and other substances, in highland barley seeds. Therefore, in order to clarify the work mechanism of the response of salt stress resistance in highland barley, the detailed analysis of the phenylpropane biosynthesis pathway was indispensable.

The first three catalytic reactions in the phenylpropane metabolic pathway were demonstrated to be the core reactions of the whole metabolic pathway [41]. PAL is the first reactive enzyme in the phenylpropane metabolism pathway, and also a rate-limiting enzyme in catalyzing phenylpropane to cinnamic acid, which is the precursor for the synthesis of downstream secondary metabolites [42]. Sang [43] found that the PAL activity was increased in wheat roots

with the increase of $Na^+$, especially in the salt-sensitive wheat genotypes. This study found that the transcription of three PAL genes (a total of 9 PAL genes) was significantly up-regulated under salt stress conditions, particularly D1007_11280, with a maximum increase in gene transcription level ($Log_2FoldChange = 4.945$). With the coordinated expression of C4H and PAL, p-hydroxycinnamic acid can be produced by hydroxylation of the C4 position of cinnamic acid, which is the second step of phenylpropane metabolism. Baek [44] cloned C4H gene from black raspberry and found that it had a differential expression pattern during fruit development. The gene regulation of C4H could change the content of black raspberry flavonoids. In this study, we found that the transcriptional level of one of the C4H genes, D1007_58538, in highland barley seed cells was significantly up-regulated, reaching about 8 (23.003) times on the third day during germination stage, respectively (Fig 5 and S4 Table). 4CL is the last catalytic enzyme in the three standard steps of the phenylpropane metabolic pathway, and it is also the main fulcrum enzyme in the secondary metabolism of the life-generating thioester [45]. Our study found that transcription levels of two 4CL genes (a total of three) in highland barley seed cells were significantly up-regulated (S4 Table).

Several key enzymes, including CCR, CAD, COMT, and CCoAOMT, work to produce phenylpropane secondary metabolites, such as polyphenols, flavonoids, lignin and methyl cinnamate. In this study, three CAD genes, four CCR genes, five COMT genes, and one CCoAOMT gene were demonstrated to respond to salt stress (S4 Table). The results showed that transcription levels of these key enzymes reached the maximum on the third day during germination stage. Except that, the transcriptional level of each key metabolic gene, including two PAL genes, CH4 gene, 4CL gene, COMT gene, CCoAOMT gene, CCR gene, and CAD gene, was accurately analyzed through qRT-PCR method (Fig 6), which further verify that the key points of the phenylpropane metabolic pathway in highland barley seed cells could sensitively response to salt stress. It has been shown that the expression of these genes was considered as the regulatory factors controlling the phenylpropane pathway, and overexpression of these genes in highland barley would improve the resistance of highland barley to stress [46].

## Conclusion

Indeed, the resistance mechanism of highland barley cannot focus on just one metabolic pathway. For example, it was found in this study that the glutathione synthesis pathway and energy synthesis pathway of highland barley seed cells were significantly improved after salt stress. What's more, it's necessary to analyze the salt tolerance of different varieties of highland barley in the next step, and then screen out susceptible plants for further understanding of salt tolerance mechanism as well as new pathway. These issues need to be further addressed in our future studies.

## Supporting information

**S1 Fig. Phenotypic characterization of highland barley.**
(TIF)

**S2 Fig. Content analysis of highland barley seeds under different salt concentration.**
(TIF)

**S3 Fig. Volcano plot analysis of differentially expressed genes.**
(TIF)

**S4 Fig. GO enrichment analysis of differentially expressed genes.**
(TIF)

**S5 Fig. KEGG analysis of phenylpropane metabolic pathway.**
(TIF)

**S1 Table. The content of antioxidants.**
(DOCX)

**S2 Table. Characteristics of libraries.**
(DOCX)

**S3 Table. Genes altered significantly in transcription level.**
(DOCX)

**S4 Table. Transcriptional level analysis of key genes in the phenylpropane metabolic pathway.**
(DOCX)

**S5 Table. Primers used for qRT-PCR.**
(DOCX)

## Acknowledgments

We gratefully acknowledge the Academy of Agriculture and Animal Husbandry Sciences of Tibet Autonomous Region provided Tibet Qing 2000 seeds.

## Author Contributions

**Conceptualization:** ZhengLian Xue, BingSheng Wang, MengDie Tao, Ming Zhao.

**Data curation:** GuoQiang Zhang, ShiGuang Zhao.

**Funding acquisition:** ZhengLian Xue, Ming Zhao, ShiGuang Zhao.

**Investigation:** ShiGuang Zhao.

**Methodology:** BingSheng Wang, Ming Zhao.

**Project administration:** ZhengLian Xue, Ming Zhao, ShiGuang Zhao.

**Resources:** ShiGuang Zhao.

**Software:** BingSheng Wang, ChangYu Qu, MengDie Tao, ShiGuang Zhao.

**Supervision:** ShiGuang Zhao.

**Validation:** BingSheng Wang, Zhou Wang, Ming Zhao.

**Visualization:** BingSheng Wang.

**Writing – original draft:** BingSheng Wang, Ming Zhao.

**Writing – review & editing:** Ming Zhao, ShiGuang Zhao.

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
