## [Decision Letter · Decision Letter 0]

30 Mar 2023

PONE-D-23-04873Response of stress resistance in highland barley ( Hordeum vulgare L. var. nudum ) through phenylpropane metabolic pathwayPLOS ONE

Dear Dr. Zhao,

Thank you for submitting your manuscript to PLOS ONE. After careful consideration, we feel that it has merit but does not fully meet PLOS ONE’s publication criteria as it currently stands. Therefore, we invite you to submit a revised version of the manuscript that addresses the points raised during the review process.

ACADEMIC EDITOR: Dear Authors,

Three independent expert reviewers have finalized their revisions to the manuscript.

Based on their comments, the manuscript cannot be accepted in its present form but needs extensive revisions.

The main criticisms regard the Results and Discussion sections that should be rechecked before accepting.

The Discussion must be facilitated with an adequate discussion of results avoiding speculation based on previous studies.

The English language needs significant revision to meet the journal's standards.

I encourage the authors to proceed with revisions and submit the revised version of the manuscript.

We look forward to receiving your revised manuscript.

Kind regards,

Mojtaba Kordrostami, Ph.D.

Academic Editor

PLOS ONE

Journal Requirements:

Reviewers' comments:

Reviewer's Responses to Questions

**Comments to the Author**

1. Is the manuscript technically sound, and do the data support the conclusions?

Reviewer #1: Yes

Reviewer #2: Partly

Reviewer #3: Partly

2. Has the statistical analysis been performed appropriately and rigorously? 

Reviewer #1: Yes

Reviewer #2: No

Reviewer #3: Yes

3. Have the authors made all data underlying the findings in their manuscript fully available?

Reviewer #1: Yes

Reviewer #2: Yes

Reviewer #3: Yes

4. Is the manuscript presented in an intelligible fashion and written in standard English?

Reviewer #1: Yes

Reviewer #2: Yes

Reviewer #3: Yes

5. Review Comments to the Author

Reviewer #1: Why was gene expression of more important products of the phenylpropane pathway such as chavicol and methyl cinnamate not investigated in this research?

What was the temperature of the intelligent artificial climate Chamber during the germination of the seeds?

How many seeds were placed in each seedling boxes?

Why was NaCLO used?

Why was polyphenolic or percolation method not used for extracting?

According to the results of the research, the discussion section should be expanded to better show the different aspects of the research topic.

Reviewer #2: The manuscript "Response of stress resistance in highland barley ( Hordeum vulgare L. var. nudum ) through phenylpropane metabolic pathway" has been written well but needs minor revisions.

The corrections has been highlighted in the manuscript file, please follow.

The need to give ANOVA for significance of studied phenotypic traits.

Reviewer #3: In title: type of stress? is not addressed.

In introduction:

in the end paragraph, authors findings for instance "Interestingly, the phenylpropane metabolic pathway was the most important pathway associated with the stress resistance of highland barley………" were represented that must transfer to section results.

In materials and methods:

-Recommended to use susceptible genotype for further understanding of salt tolerance mechanism as well as new pathway if there is.

- The reason for selection of salinity level of 240 mmol/L? emphasizing on previous reports is not sufficient. Is there any data acquired by authors in this field?

In Results:

Number of biological replications related to RNA-seq experiment?

6. PLOS authors have the option to publish the peer review history of their article (what does this mean?). If published, this will include your full peer review and any attached files.

Reviewer #1: No

Reviewer #2: **Yes: **Dr. Qurban Ali

Reviewer #3: No

---

## [Author Response · Author response to Decision Letter 0]

17 May 2023

Editor's Comments to Author:

Comment 1: Please ensure that your manuscript meets PLOS ONE's style requirements, including those for file naming. 

Answer 1: Thank you very much for your suggestions. We have revised our manuscript according to the PLOS ONE's style requirements.

Comment 2: In your Data Availability statement, you have not specified where the minimal data set underlying the results described in your manuscript can be found. 

Answer 2: Thank you very much for your suggestions. We apologize for the lack of information for this part. We have supplemented the information of our study’s minimal underlying data (please see lines 218-220 and Table S2) in the revised-manuscript.

Reviewer #1

Comment 1: Why was gene expression of more important products of the phenylpropane pathway such as chavicol and methyl cinnamate not investigated in this research?

Answer 1: Thank you very much for your comment. Because of that phenylpropane metabolic pathway products such as polyphenols, flavonoids, and lignin play an important role in enhancing plant stress resistance, the above product synthesis pathways were just investigated in this research. KEGG enrichment analysis also showed that DEGs in highland barley cells under salt stress were significantly enriched in the above product synthesis pathways. Certainly, lacking of informations for other products of the phenylpropane pathway could not make the data fully the conclusions. We apologize for the lack of results for this part. Therefore, we have added the methyl cinnamate synthesis pathway in this research and screened its DEG (Figure 5 and Figure 6). The expression level of related enzymes in chavicol synthesis pathway did not significantly change under salt stress. We have added the discussion (please see lines 298) in the revised-manuscript.

Comment 2: What was the temperature of the intelligent artificial climate Chamber during the germination of the seeds?

Answer 2: Thank you very much for your comment. We apologize for the lack of information for this part. We choose 25℃ as the culture temperature of the intelligent artificial climate chamber during the germination of the seeds, because the highland barley can tolerate cold in the whole growth period, and generally can grow well in the area where the annual average temperature is above 3℃. It was found that the germination rate, germination potential and germination index of highland barley seeds reached the highest at 25℃. We have added this information (please see lines 98) in the revised-manuscript.

Comment 3: How many seeds were placed in each seedling boxes?

Answer 3: Thank you very much for your comment. We apologize for the lack of information for this part. About 20 g of seeds are placed in each box to ensure that there is space between each seed and that it does not pile up, thus ensuring germination rate and growth. We have added this information (please see lines 96-97) in the revised-manuscript.

Comment 4: Why was NaCLO used

Answer 4: Thank you very much for your comment. As a typical example of growing in extreme environmental conditions, highland barley has its unique tolerance to adversity and is very suitable for growing on saline soil as a food crop. We analyzed the response mechanism of barley under NaCl condition by transcriptomics and qRT-PCR, which laid the foundation for improving the utilization rate of saline soil. We have added the discussion (please see lines 98-100) in the revised-manuscript.

Comment 5: Why was polyphenolic or percolation method not used for extracting?

Answer 5: Thank you very much for your comment. The ultrasonic assisted extraction method adopted in this experiment is to extract the effective components in highland barley by increasing the motion velocity and penetration of medium molecules by using the mechanical effect, cavitation effect and thermal effect of ultrasonic. Ultrasonic wave can dissolve polyphenols, flavonoids and other substances contained in plants in methanol as completely as possible. The method has the advantages of short time, low temperature, wide adaptability, less impurity and simple extraction. We have added the Preparation of Highland Barley Extract Solution (please see lines 103-105) in the revised-manuscript.

Comment 6: According to the results of the research, the discussion section should be expanded to better show the different aspects of the research topic

Answer 6: Thank you very much for your comment. We apologize for the imperfect discussiion section in our manuscript. We have rewritten the discussiion section and the revised parts were marked in red in the revised-manuscript.

Reviewer #2

Comment 1: The corrections has been highlighted in the manuscript file, please follow.

Answer 1: Thank you very much for reviewing this manuscript and revising the questions. Revised portions were marked in red in the revised manuscript.

Comment 2: The need to give ANOVA for significance of studied phenotypic traits.

Answer 2: Thank you very much for your comment. We have supplement ANOVA for significance of the content of antioxidants.We have added these results (please see lines 185 and Table S1) in the revised-Supplementary information.

Table S1 The content of antioxidants

Day Content of total phenolic（mg·g-1DW） Content of total flavonoid（mg·g-1DW） Content of total ferulic acid（μg·g-1DW） Content of total caffeic acid（μg·g-1DW）

 non-salt salt non-salt salt non-salt salt non-salt salt

0d 2.15±0.25d 3.46±0.26e 0.98±0.11c 1.01±0.01d 2.01±0.14e 2.35±0.05e 0.21±0.034e 0.26±0.06e

1d 2.76±0.21b 5.53±0.47d 1.01±0.13c 1.05±0.02d 3.83±0.12c 7.31±0.06c 1.26±0.15d 1.61±0.19d

3d 3.09±0.26a 8.38±0.36a 1.62±0.15a 1.98±0.08a 8.97±0.06a 12.22±0.35d 2.89±0.16a 4.69±0.24a

5d 2.31±0.14c 7.97±0.23b 1.20±0.11b 1.48±0.04b 4.59±0.14b 8.53±0.23b 1.98±0.14b 2.71±0.14b

7d 2.19±0.18cd 7.38±0.28c 1.02±0.09c 1.20±0.05c 2.94±0.1d 4.34±0.32d 1.42±0.07c 2.12±0.05c

Note: The data are the mean ± standard deviation of 3 independent experiments

There was statistical significance for different characters (p < 0.05).

Reviewer #3

Comment 1: In title: type of stress? is not addressed

Answer 1:  Thank you very much for your comment. We have revised the title (please see lines 2) in the revised-manuscript.

Comment 2: In introduction:in the end paragraph, authors findings for instance "Interestingly, the phenylpropane metabolic pathway was the most important pathway associated with the stress resistance of highland barley………" were represented that must transfer to section results.

Answer 2:  Thank you very much for your advice. We have transferred "Interestingly, the phenylpropane metabolic pathway was the most important pathway associated with the stress resistance of highland barley………" from the introduction to the section results (please see lines 286-290) in the revised-manuscript.

Comment 3: Recommended to use susceptible genotype for further understanding of salt tolerance mechanism as well as new pathway if there is.

Answer 3:  Thank you very much for your suggestion.We plan to analyze the salt tolerance of different varieties of highland barley in the next step, and then screen out susceptible plants for further understanding of salt tolerance mechanism as well as new pathway(please see lines 405-408).

Comment 4: The reason for selection of salinity level of 240 mmol/L? emphasizing on previous reports is not sufficient. Is there any data acquired by authors in this field?

Answer 4: Thank you very much for your comment. We previously determined the contents of total polyphenols and total flavonoids in highland barley under different NaCl concentrations, and found that the contents of total polyphenols and flavonoids reached the maximum under the condition of 240 mmol/L. Therefore, 240 mmol/L NaCl was selected as the salt stress concentration. We have added these results (please see lines 181-183 and Figure S2) in the revised-Supplementary information.

Figure S2. Content analysis of highland barley seeds under different salt concentration. (A) The content of total polyphenols under different salt concentration. (B) The contents of total flavonoids changes under different salt concentration

Comment 5: Number of biological replications related to RNA-seq experiment?

Answer 4: Thank you very much for your question. We apologize for the lack of information for this part. There were 3 biological replicates for each group of RNA-seq data. We have added this information (please see lines 156) in the revised-manuscript.

---

## [Decision Letter · Decision Letter 1]

26 May 2023

Response of salt stress resistance in highland barley ( Hordeum vulgare L. var. nudum ) through phenylpropane metabolic pathway

PONE-D-23-04873R1

Dear Dr. Zhao,

We’re pleased to inform you that your manuscript has been judged scientifically suitable for publication and will be formally accepted for publication once it meets all outstanding technical requirements.

Kind regards,

Mojtaba Kordrostami, Ph.D.

Academic Editor

PLOS ONE

Additional Editor Comments (optional):

Reviewers' comments:

Reviewer's Responses to Questions

**Comments to the Author**

1. If the authors have adequately addressed your comments raised in a previous round of review and you feel that this manuscript is now acceptable for publication, you may indicate that here to bypass the “Comments to the Author” section, enter your conflict of interest statement in the “Confidential to Editor” section, and submit your "Accept" recommendation.

Reviewer #2: All comments have been addressed

Reviewer #3: All comments have been addressed

2. Is the manuscript technically sound, and do the data support the conclusions?

Reviewer #2: Yes

Reviewer #3: Partly

3. Has the statistical analysis been performed appropriately and rigorously? 

Reviewer #2: Yes

Reviewer #3: Yes

4. Have the authors made all data underlying the findings in their manuscript fully available?

Reviewer #2: Yes

Reviewer #3: Yes

5. Is the manuscript presented in an intelligible fashion and written in standard English?

Reviewer #2: Yes

Reviewer #3: Yes

6. Review Comments to the Author

Reviewer #2: (No Response)

Reviewer #3: Dear chief editor

Based on reviewing results, the manuscript can be published in this journal.

Best regards

7. PLOS authors have the option to publish the peer review history of their article (what does this mean?). If published, this will include your full peer review and any attached files.

Reviewer #2: No

Reviewer #3: **Yes: **Hamid Hatami Maleki

---

## [Editor Report · Acceptance letter]

2 Jun 2023

PONE-D-23-04873R1 

Response of salt stress resistance in highland barley (*Hordeum vulgare L. var. nudum*) through phenylpropane metabolic pathway 

Dear Dr. Zhao:

I'm pleased to inform you that your manuscript has been deemed suitable for publication in PLOS ONE. Congratulations! Your manuscript is now with our production department. 

Kind regards, 

on behalf of

Dr. Mojtaba Kordrostami 

Academic Editor

PLOS ONE